# Optimal Shadowing Filter for a Positioning and Tracking Methodology with Limited Information

**DOI:** 10.3390/s19040931

**Published:** 2019-02-22

**Authors:** Ayham Zaitouny, Thomas Stemler, Shannon Dee Algar

**Affiliations:** 1Department of Mathematics and Statistics, University of Western Australia, 35 Stirling Highway, Crawley, WA 6009, Australia; thomas.stemler@uwa.edu.au (T.S.); shannon.algar@research.uwa.edu.au (S.D.A.); 2Commonwealth Scientific and Industrial Research Organisation, 26 Dick Perry Ave, Kensington, WA 6151, Australia

**Keywords:** shadowing filter, tracking, positioning, correlated noise, irregular sampling, singularities

## Abstract

Positioning and tracking a moving target from limited positional information is a frequently-encountered problem. For given noisy observations of the target’s position, one wants to estimate the true trajectory and reconstruct the full phase space including velocity and acceleration. The shadowing filter offers a robust methodology to achieve such an estimation and reconstruction. Here, we highlight and validate important merits of this methodology for real-life applications. In particular, we explore the filter’s performance when dealing with correlated or uncorrelated noise, irregular sampling in time and how it can be optimised even when the true dynamics of the system are not known.

## 1. Introduction

A wide range of filters has been developed for the purposes of tracking moving objects and approximating their trajectories. Currently, tracking is typically achieved using sequential statistical filters, such as Kalman or particle filters, which take only the current and future states into account [1,2,3,4,5,6,7]. The classic Kalman filter has been extended to consider converted measurements and nonlinear problems [8]. A review of how to use the particle filter and its capability to handle general models and applications has been introduced in [9,10]. In [11,12,13], we developed and introduced a new tracking methodology based on shadowing filters. We proved the concept and demonstrated the method’s superiority over other tracking methods, as well as its applicability to real-life problems. In this paper, we conduct further investigations to demonstrate and verify the additional advantages of the shadowing filter tracking method, which are important for real applications. Namely, we introduce the two-dimensional solution when the observational error’s correlation is taken into account and compare the method’s performance when this correlation is considered or ignored. We additionally test the filter’s performance when dealing with non-uniform sampling in time and demonstrate the filter’s capability to overcome singularities.

As a motivation, consider the frequently-occurring tracking problem that arises when observations are recorded in a non-Cartesian coordinate system. Assume that the bearing and range of an object are observed. These observations are independently determined from some reference point. However, when these measurements are transformed into Cartesian data, they will have correlated errors resulting from this transformation. To make matters worse, we often need to combine multiple measurements. In practice, such problems arise in: (a) active radar locations; (b) satellite interferometry; (c) GPS; and (d) passive sonar locations. The challenges in these applications vary. Applications (a), (b) and (d) all use bearing and range data, but in (b), the range is very accurately measured, while the bearing has large errors. In (d), the situation is the opposite, with observations of the bearing being high quality. Application (c), on the other hand, uses only range information, but from several different satellites. All of these cases can be dealt with using the shadowing filter by taking into account the correlation of the observational error. Section 3 of this paper introduces the methodology for approximating the trajectory of an object being tracked with noisy position data and potential correlations in the observational error. The solution in two-dimensions is derived for both uncorrelated and correlated error scenarios. Section 4 utilizes a simple, non-trivial system example to perform a numerical analysis in order to assess whether or not error correlation has to be considered to ensure a good approximation of the object’s path and if the extra computational effort required is worth it.

Other common challenges one could face in practice are singularities and non-uniform sampling of data caused by the failure of the recording devices. Section 5 demonstrates the robustness of the shadowing filter tracking methodology to avoid singularity impacts, as well as deal successfully with irregular time resolution. We also apply the tracking technique to a noisy chaotic trajectory, specifically that of the Lorenz model, and analyse the results from the perspective of noise reduction. In the final section, we draw conclusions and make some suggestions for future directions of this work.

## 2. Shadowing Filters

The shadowing filter method is unlike sequential filter methods in that it estimates the full trajectory based on all observations simultaneously. In this aspect, our approach is somewhat similar to variational filters [14,15]; however, it does not fall into the trap of local minima [16,17]. This is due to the very simple principle these filters are based on: if the model is good enough, state estimations must be close to the observations and consistent with the model’s equations. In practice, this principle imposes a quadratic norm on the filter, which only has one minimum. Specifically regarding the tracking approach, previous investigations showed and verified some merits and advantages of the shadowing filter tracking methodology. In [12,13], we showed the capability of shadowing filter to track single-particle and multi-particle moving objects using GPS data from flying birds; in Figure 1, we show an example of how we successfully used this method to track eight pigeons’ trajectories and extracted the corresponding acceleration profiles only from positional GPS data. To track rigid bodies, the method was extended [11] to include rotational motion and moments of inertia. Our method is able to reconstruct the full dynamical phase space from positional information [11,12,13].

In general, varieties of the shadowing filter have proven to have better performance than Kalman filters [18], particle filters [19], sliding average filters [12] and variational filters [17]. They have been successfully implemented from simple low-dimensional maps and flows [17] up to operational weather models [20]. In [11], we addressed a benchmark with the state-of-art of tracking methods. We conducted direct detailed comparisons with the Kalman filter [21], extended Kalman filter [22] and particle filter [23,24] approaches. Our computational comparisons confirm the superiority of the shadowing filter tracking methodology over these sequential filter tracking methods with respect to accuracy and complexity (computational time). In this manuscript, we will not redo these comprehensive numerical comparisons. However, as the motivation, we confirm the performance compared to existing methods in Figure 2 with systems later investigated in this manuscript. Specifically, correlated observational errors are taken into account in Figure 2a,b and when singularities arise due to recording failure in Figure 2c,d. More detailed explanations and investigations about these scenarios and others are addressed in the following sections.

## 3. Summary of the Tracking Methodology

Typically, tracking data will contain only noisy position observations, and so, we are faced with the challenge of having no explicit information about the object’s acceleration or velocity. Using the acceleration as a free parameter, we are able to implement the shadowing filter to find a good approximation of the full phase space including acceleration and velocity information based on the position observations. This free parameter can then be tuned in order to optimize the quality of the solution of the approximated states.

We aim to track a point object that is moving in a *d*-dimensional Euclidean space with Cartesian coordinates, given a sequence of noisy observations. Let $R_{i}\in R^{d}$ be the true states and $P_{i}\in R^{d}$ be the noisy observations of the position at time $t_{i}$ for i=0,1,…,n, where the observational errors have a d×d covariance matrix $C_{i}$ and corresponding information matrix $I_{i}=C_{i}^{-1}$, which is used to account for the correlations of the observational error.

The object’s dynamics are modelled based on its observed position $P_{i}\in R^{d}$, velocity $v_{i}\in R^{d}$ and acceleration $a_{i}\in R^{d}$ for $t_{i}\leq t\leq t_{i+1}$. Our aim is to approximate $p_{i}\in R^{d}$ close to the true state $R_{i}$. To do this, we minimize the total squared error $\sum_{i=0}^{n}{\left(P_{i}-p_{i}\right)}^{T}I_{i}\left(P_{i}-p_{i}\right)$.

We assume that the acceleration is constant over one time interval $\left(T_{i}=t_{i+1}-t_{i}\right)$ and that its magnitude is bounded over the entire trajectory by the relation $\left(\sum_{i=0}^{n-1}T_{i}a_{i}^{T}a_{i}\leq \left(t_{n}-t_{0}\right)\xi^{2}\right)$. Using Newton’s laws and a Galilean transformation, we can solve this optimization problem with the Lagrange multipliers method [25]. The appropriate Lagrange function is:
(1a)$$\begin{array}{ccc} L= & \frac{1}{2} & \underset{i=0}{\sum^{n}}{\left(P_{i}-p_{i}\right)}^{T}I_{i}\left(P_{i}-p_{i}\right) \end{array}$$
(1b)$$\begin{array}{cc} + & \underset{i=0}{\sum^{n-1}}\lambda_{i+1}^{T}\left(p_{i+1}-p_{i}-v_{i}T_{i}-\frac{1}{2}a_{i}T_{i}^{2}\right) \end{array}$$
(1c)$$\begin{array}{cc} + & \underset{i=0}{\sum^{n-1}}\mu_{i+1}^{T}\left(v_{i+1}-v_{i}-a_{i}T_{i}\right) \end{array}$$
(1d)$$\begin{array}{cc} + & \eta (\underset{i=0}{\sum^{n-1}}T_{i}a_{i}^{T}a_{i}-\left(t_{n}-t_{0}\right)\xi^{2}), \end{array}$$
where the Lagrange multipliers $\lambda_{i}\in R^{d},\mu_{i}\in R^{d}$ are *d*-dimensional column vectors, η∈R and ξ∈R. Equation (1b) represents Newton’s first law, Equation (1c) Newton’s second law, while Equation (1d) bounds the magnitude of the acceleration. While using the Lagrange multiplier method seems only natural in the context of a Newtonian model, the optimization could be also implemented using for example gradient descent [16,17] or penalty methods [26].

The solution to the scalar case and rigid-body problems can be found in [11,27]. In this paper, using similar linear algebra methods and appropriately-constructed vector and matrix equations, we extend the solution to higher dimensions, specifically two-dimensional problems, taking into account the correlation of the observational errors. Note that this solution can be generalised for the *d*-dimensional problem where all vectors and matrices increase in dimension by a factor of *d*.

### 3.1. Solution of the Two-Dimensional Problem

#### 3.1.1. Correlated Observational Errors

For higher dimensions, even in two dimensions, the complexity increases as we have to determine whether observational errors are not independent, but correlated. Note that the Weighted Least Squares (WLS) method and its extension, the constrained-WLS, are related to our shadowing filter [28,29,30,31]. For the tracking problem, they are able to deal with correlated/uncorrelated noise and irregular sampling in time. However, unlike the shadowing filter, they do not consider explicitly the velocity and acceleration [32,33].

Consider the situation of an object moving in the xy-plane. Given a time resolution of measurements $T_{i}=t_{i+1}-t_{i}$ for i=0,1,…,n−1, the true position is $R_{i}={\left(R_{x_{i}},R_{y_{i}}\right)}^{T}\in R^{2}$ and the noisy observed position is $P_{i}={\left(X_{i},Y_{i}\right)}^{T}\in R^{2}$. The error in the *x*-direction has variance $\sigma_{x_{i}}^{2}$, and similarly, in the *y*-direction the error has a variance $\sigma_{y_{i}}^{2}$. Therefore, the observational error has the following 2×2 covariance matrix:$$C_{i}=\left(\begin{array}{cc} \sigma_{x_{i}}^{2} & {\mathrm{cov}}_{x_{i}y_{i}}\\ {\mathrm{cov}}_{y_{i}x_{i}} & \sigma_{y_{i}}^{2} \end{array}\right),$$
with the corresponding information matrix:$$I_{i}=C_{i}^{-1}=\left(\begin{array}{cc} S_{x_{i}} & S_{x_{i}y_{i}}\\ S_{x_{i}y_{i}} & S_{y_{i}} \end{array}\right),$$
where:$$\begin{array}{ccc} S_{x_{i}} & = & \frac{\sigma_{y_{i}}^{2}}{\sigma_{x_{i}}^{2}\sigma_{y_{i}}^{2}-{\mathrm{cov}}_{x_{i}y_{i}}^{2}},\\ S_{y_{i}} & = & \frac{\sigma_{x_{i}}^{2}}{\sigma_{x_{i}}^{2}\sigma_{y_{i}}^{2}-{\mathrm{cov}}_{x_{i}y_{i}}^{2}},\\ S_{x_{i}y_{i}} & = & \frac{-{\mathrm{cov}}_{x_{i}y_{i}}}{\sigma_{x_{i}}^{2}\sigma_{y_{i}}^{2}-{\mathrm{cov}}_{x_{i}y_{i}}^{2}}. \end{array}$$

Note that generally, $\sigma_{x_{i}}$ and $\sigma_{y_{i}}$ can vary with time.

In addition, defining the unknown variables, $p_{i}={\left(x_{i},y_{i}\right)}^{T}\in R^{2}$, $v_{i}={\left(v_{x_{i}},v_{y_{i}}\right)}^{T}\in R^{2}$, $a_{i}={\left(a_{x_{i}},a_{y_{i}}\right)}^{T}\in R^{2}$, $\lambda_{i}={\left(\lambda_{x_{i}},\lambda_{y_{i}}\right)}^{T}\in R^{2}$, $\mu_{i}={\left(\mu_{x_{i}},\mu_{y_{i}}\right)}^{T}\in R^{2}$, η∈R and ξ∈R, then the Lagrangian expressed by Equation (1) can be stated as:
(2a)$$\begin{array}{ccc} L= & \frac{1}{2} & \sum_{i=0}^{n}S_{x_{i}}{\left(X_{i}-x_{i}\right)}^{2}+2S_{x_{i}y_{i}}\left(X_{i}-x_{i}\right)\left(Y_{i}-y_{i}\right)+S_{y_{i}}{\left(Y_{i}-y_{i}\right)}^{2}\\ & + & \sum_{i=0}^{n-1}[\lambda_{x_{i+1}}\left(x_{i+1}-x_{i}-v_{x_{i}}T_{i}-\frac{1}{2}a_{x_{i}}T_{i}^{2}\right) \end{array}$$
(2b)$$\begin{array}{c} +\lambda_{y_{i+1}}\left(y_{i+1}-y_{i}-v_{y_{i}}T_{i}-\frac{1}{2}a_{y_{i}}T_{i}^{2}\right)] \end{array}$$
(2c)$$\begin{array}{cc} + & \sum_{i=0}^{n-1}\mu_{x_{i+1}}\left(v_{x_{i+1}}-v_{x_{i}}-a_{x_{i}}T_{i}\right)+\mu_{y_{i+1}}\left(v_{y_{i+1}}-v_{y_{i}}-a_{y_{i}}T_{i}\right) \end{array}$$
(2d)$$\begin{array}{cc} + & \eta (\sum_{i=0}^{n-1}T_{i}\left(a_{x_{i}}^{2}-a_{y_{i}}^{2}\right)-\left(t_{n}-t_{0}\right)\xi^{2}). \end{array}$$

Obviously, the optimal solution occurs when all the partial derivatives of *L* are zero:$$\frac{\partial L}{\partial x_{i}}=\frac{\partial L}{\partial y_{i}}=\frac{\partial L}{\partial v_{x_{i}}}=\frac{\partial L}{\partial v_{y_{i}}}=\frac{\partial L}{\partial a_{x_{i}}}=\frac{\partial L}{\partial a_{y_{i}}}=\frac{\partial L}{\partial \lambda_{x_{i}}}=\frac{\partial L}{\partial \lambda_{y_{i}}}=\frac{\partial L}{\partial \mu_{x_{i}}}=\frac{\partial L}{\partial \mu_{y_{i}}}=\frac{\partial L}{\partial \eta }=0.$$

The partial derivative equations and the complete solution of this situation are detailed in Appendix A. For the purpose of approximating the trajectory, we require only the final equation of the solution given by Equation (A24):$$\left(\eta \bar{B}+\bar{A}I\right)p=\bar{A}IP$$
where $\bar{A}$ and $\bar{B}$ are numerical matrices used to construct the solution, I is the information matrix, P is the observational states vector, p is the approximated states vector and η is the smoothing parameter of the filter.

#### 3.1.2. Uncorrelated Observational Errors

If the observational errors of the *x*-component are independent of those in the *y*-component, then they are uncorrelated and at each time $t_{i}$ where i=0,1,…,n the covariance between $x_{i}$ and $y_{i}$ equals zero, i.e., ${\mathrm{cov}}_{x_{i}y_{i}}=0,\;\forall i=0,1,\ldots ,n$. Therefore, $S_{x_{i}}=\sigma_{x_{i}}^{-2}$, $S_{y_{i}}=\sigma_{y_{i}}^{-2}$, $S_{x_{i}y_{i}}=0$. Consequently, Equations (A1) and (A2) can be stated as: (3)$$\begin{array}{ccc} \frac{\partial L}{\partial x_{i}} & = & 0=\left\{\begin{array}{cc} -\sigma_{x_{0}}^{-2}\left(X_{0}-x_{0}\right)-\lambda_{x_{1}},\; & i=0\\ -\sigma_{x_{i}}^{-2}\left(X_{i}-x_{i}\right)+\lambda_{x_{i}}-\lambda_{x_{i+1}},\; & 0<i<n\\ -\sigma_{x_{n}}^{-2}\left(X_{n}-x_{n}\right)+\lambda_{x_{n}},\; & i=n \end{array}\right. \end{array}$$
(4)$$\begin{array}{ccc} \frac{\partial L}{\partial y_{i}} & = & 0=\left\{\begin{array}{cc} -\sigma_{y_{0}}^{-2}\left(Y_{0}-y_{0}\right)-\lambda_{y_{1}},\; & i=0\\ -\sigma_{y_{i}}^{-2}\left(Y_{i}-y_{i}\right)+\lambda_{y_{i}}-\lambda_{y_{i+1}},\; & 0<i<n\\ -\sigma_{y_{n}}^{-2}\left(Y_{n}-y_{n}\right)+\lambda_{y_{n}},\; & i=n \end{array}\right. \end{array}$$

This gives us two independent systems of equations: the *x*-component system Equations (A3), (A5), (A7), (A9) and (3) and the *y*-component system Equations (A4), (A6), (A8), (A10) and (4). For each component, the systems are identical to the scalar case, which has been solved previously [11,27], and the optimization problem can be solved independently for *x* and *y*.

## 4. Numerical Investigations: Correlated vs. Uncorrelated Observational Errors

Given a dataset with correlated observational errors, we can either take the correlation into account or ignore it. There are practical reasons why one might choose to ignore the correlation. For instance, the algorithms are more complex with correlation and much more computationally intensive, as a d×5-dimensional system of equations must be solved (see Equations (A1)–(A10)), but if we ignore correlation, the problem is reduced to *d* independent scalar problems.

We refer herein to these two situations, of taking the Correlation into Account and Ignoring it, as CA and CI, respectively.

In the following investigations, a target’s position is tracked in the xy-plane by the position $s={\left(x,y\right)}^{T}$ given by the simple, non-trivial observation model:(5)s=xy=10(t/150−1/2)10(t/150−1/2)−(5/2)sin(2π(t/150−1/2)),
where 0≤t≤150, i.e., the length of the time series is n=151.

Note that the choice of this particular path is motivated by the range and bearing problem set up below with the non-linearity in *y* simplifying the description of the dynamics in polar coordinates.

We introduce correlated observational errors by transforming the position into range and bearing coordinates $q={\left(r,\theta \right)}^{T}$ with the inverse of the standard transformation s=f(q):(6)xy=a+rcosθb+rsinθ,
where *r* is the range from a reference point ${\left(a,b\right)}^{T}$ and θ is the bearing in radians measured anti-clockwise from the *x*-axis. Thus, the transformed observations are ${\left(r_{i},\theta_{i}\right)}^{T}$, where i=0,1,…,150 and the origin ${\left(0,0\right)}^{T}$ is the reference point.

We then add white noise to each measurement:RiΘi=ri+σrχiθi+σθψi,
where $\chi_{i}∼N\left(0,1\right)$ is the observational noise of the range component with variance $\sigma_{r}^{2}$ and $\psi_{i}∼N\left(0,1\right)$ is the observational noise of the bearing component with variance $\sigma_{\theta }^{2}$.

Finally, we obtain the noisy observations, $P_{i}={\left(X_{i},Y_{i}\right)}^{T}$, in Cartesian coordinates, by applying the transformation defined in Equation (6) to ${\left(R_{i},\Theta_{i}\right)}^{T}$.

Under the assumption that *r* is not close to zero and the variances of *r* and θ are small, the information matrix of $s={\left(x,y\right)}^{T}$ can be approximated as [27]:$$I_{s}=\left(\begin{array}{cc} \sigma_{r}^{-2}\cos^{2}\theta +\left(1/r^{2}\right)\sigma_{\theta }^{-2}\sin^{2}\theta & \left(1/2\right)\sigma_{r}^{-2}\sin2\theta -\left(1/2r^{2}\right)\sigma_{\theta }^{-2}\sin2\theta \\ \left(1/2\right)\sigma_{r}^{-2}\sin2\theta -\left(1/2r^{2}\right)\sigma_{\theta }^{-2}\sin2\theta & \sigma_{r}^{-2}\sin^{2}\theta +\left(1/r^{2}\right)\sigma_{\theta }^{-2}\cos^{2}\theta \end{array}\right).$$

Given that this information matrix is estimated in advance, we are now able to apply the shadowing technique via Equation (A24) to approximate the trajectory, which we illustrate in Figure 3.

In Figure 3a, the range measurement is four-times more accurate than that of the bearing, as, for this experiment, the standard deviations were chosen to be $\sigma_{\theta }=0.60$ and $\sigma_{r}=0.15$. Conversely, in Figure 3b, the bearing measurement is four-times more accurate than the range, as $\sigma_{r}=0.60$ and $\sigma_{\theta }=0.15$. Whilst the filter does work well in either case, note that $\sigma_{\theta }$ has a significant effect on the observations. This is not the case for $\sigma_{r}$.

### 4.1. Quality Measures

In order to assess the quality of the filter, we must specify the aim of the tracking. Our aim is to find an approximate trajectory from noisy observations of a moving object’s positions, which lies optimally close to the true trajectory of the object. In this context, there are two questions that one might hope to answer: (1) Where is the object in a given time interval? (2) What is the current position?

The different nature of these questions is subtle, but profound, with each lending itself to a different class of possible applications. For Question (1), we need to find an approximate trajectory that is close to the true trajectory over the entire interval and therefore shadows the true dynamics. Such a scenario would be required, for example, in order to identify the ship responsible for leaking oil into the ocean based on its trajectory. Question (2) does not need an approximated trajectory that is close to all states, but only close to the last state, for example finding a missing flight. A filter capable of solving both questions will likely need to be optimized differently for the particular case. This could be either “tracking the path that the object travelled” or “knowing the recent position of the object”. Consequently, we require two measures.

The first measure we define details how close the approximated states $p_{i}={\left(x_{i},y_{i}\right)}^{T}$ are to the true states $R_{i}={\left(R_{x_{i}},R_{y_{i}}\right)}^{T}$ along the whole trajectory for i=0,1,…,n where n+1 is the length of the time series. For this, we use the root mean squared error defined by:(7)$$E=\sqrt{\frac{1}{1+n}\sum_{i=0}^{n}\left(∥R_{x_{i}}-x_{i}{∥}^{2}+∥R_{y_{i}}-y_{i}{∥}^{2}\right)}.$$

The second measure details how close the approximated final state, $p_{n}={\left(x_{n},y_{n}\right)}^{T}$, is to the true final state, $R_{n}={\left(R_{x_{n}},R_{y_{n}}\right)}^{T}$ and can be quantified by the end-point error:(8)$$e=\sqrt{∥R_{x_{n}}-x_{n}{∥}^{2}+∥R_{y_{n}}-y_{n}{∥}^{2}}.$$

For more accurate results, we use the ensemble averages of these quantities as measures of the filter’s performance. We consider *N* time series, giving *N* true trajectories, *N* sequences of observations and therefore *N* sequences of approximates:(9)$$\begin{array}{ccc} \langle E\rangle & = & \frac{1}{N}\sum_{j=1}^{N}E_{j}, \end{array}$$
(10)$$\begin{array}{ccc} \langle e\rangle & = & \frac{1}{N}\sum_{j=1}^{N}e_{j}. \end{array}$$

In the experiments that follow, we assess the quality of the filter for both tracking objectives and optimize parameters such that the respective average errors are minimized.

### 4.2. Optimization of the Filter Performance

The quality of the solution, i.e., the approximated states, can be optimized by tuning the free parameter of our filter. Generally, the dataset will contain noisy position observations, and so, acceleration can be used as the free parameter. Thus, the smoothing parameter η of the acceleration Lagrangian term in Equation (2d) approximates a suitable value of the object’s inertia.

The optimal value is dependent on the time resolution of the time series, the system’s dynamics, the noise intensity, etc. Therefore, the following investigation should be understood as a proof of concept only, whereby an optimal value is determined for some specific application. Furthermore, as we only want to show that η does in fact have some optimum value, we can use a rather simple numerical scheme.

The algorithm implemented starts with a broad parameter sweep, η=0.1,1,10,100,1000, and calculates the average errors for each tracking objective (〈E〉 and 〈e〉) from N=100 time series. It then identifies the sub-interval in which the minimum error occurs and divides this η-interval into five new subintervals. Repeating this procedure six times, with a new time series for each narrowing of the parameter range, we get an approximation of the optimal η. This approximation is sufficient to show that an optimal value exists.

Table A1 and Table A2 show the results when optimizing η subject to minimizing 〈E〉 for the CA and CI cases, respectively. The minimum in each interval is indicated in bold. We note that the optimal values for the two types of correlation consideration lie close together, and there does not seem to be a large difference between the resulting errors.

Table A3 and Table A4 show the optimization data when our aim is to minimize 〈e〉 for the CA and CI cases, respectively. Again, minimum values in each interval are indicated in bold. Here, we note a huge difference between the approximated optimal values for the two types of correlation considered. When correlation was ignored, we were required to expand the starting interval as the best η value in Experiment 1 (the zeroth iteration) was the interval’s endpoint η=1000. Consequently, the almost optimal value of this method η=1600 was much higher than the value of η=91, found when correlation was considered. On the other hand, the average end-point error was 1.4-times larger when correlation was accounted for.

The results of this investigation are summarized in Figure 4. Blue lines show the results when correlation is accounted for, and red lines show the results when correlation is ignored. The general shape of the graphs—large values of the error for small and large η values—support our initial assumption that an optimal value of η exists. Since our primitive iterative algorithm can only be used to find an approximation of the optimal η, we cannot say anything about whether the found best value is a local or a global property. However, as was pointed out in the beginning, the purpose of this investigation was only to show that the η value can be optimized for a specific application.

### 4.3. Comparison of the CA and CI Cases

Recall that CA refers to the situation where correlation is taken into account, and CI ignores it. Surprisingly, the CI algorithm seems to lead to similar or even better estimates. To get a clearer picture, we need to understand the error distributions, as well.

Figure 5 and Figure 6 show histograms of the two error measures, 〈E〉 and 〈e〉, resulting from each correlation algorithm. We used data from 100 time series and chose the approximated optimal η values found previously.

We are not surprised that there was not much difference between the CA and CI algorithms when we were interested in minimizing 〈E〉, as seen in Figure 5. Both average errors 〈E〉 were about 0.21, and the distributions were peaked around this value. If there was any difference, it was that the percentage of errors contained in the interval [0.18,24] was about 10% higher for the CA algorithm. It is not immediately clear whether this is statistically significant, but given that both methods showed similar result, we do not think that investigating this statistical significance is important.

Recall that for the case of minimizing 〈e〉, unlike for 〈E〉, our η optimization gave us quite different optimal values depending on which algorithm was used. This situation is depicted in Figure 6. Surprisingly, the resulting distributions have much in common:For the CA algorithm: the 〈e〉-distribution decayed monotonously, and all errors were contained within the interval [0.1,1.3]; and about 60% of the errors were between [0.1,0.5].For the CI algorithm: the errors were contained within the smaller interval [0.1,0.8], and the distribution was less broad; and in the subinterval from [0.1,0.5], we found 70% of the errors.In addition, there was a peak around the average value in both cases with 〈e〉≈0.2.

We conclude that the shadowing filter’s tracking methodology was robust enough to consider or ignore the correlations of the observational errors. That is, there was not a marked improvement in the shadowing filter’s performance if we took correlations into account. This is further illustrated in Figure 7, which shows estimates for both tasks ((a) minimizing average error and (b) minimizing last point error).

## 5. Applications

In this section, we demonstrate some applications with common challenges one frequently faces in practice. Specifically, we emphasise three challenges relevant to tracking: irregular/non-uniform time resolution, records with large singularities, tracking chaotic trajectories and real-world multiple object application.

### 5.1. Non-Uniform Sampling

Here, we highlight the fact that our shadowing filter is flexible enough to be used even for non-uniformly sampled data. Often, data points are not recorded with the same sampling rate, and so, $t_{h+2}-t_{h+1}=t_{h+1}-t_{h}=\Delta t$ does not hold for all *h* in the dataset. Such a situation is quite common for example in GPS tracking, as the GPS device often fails to log one or more data points [34,35]. Our filter can be used with arbitrary values of $\Delta t_{i}$ between any two data points; however, we focus here on the sub-problem that some data points are not recorded with each $\Delta t_{i}$ a multiple of some sampling time such that $\Delta t_{i}=m\times \Delta t$.

It has been shown that our tracking methodology is sufficiently robust to ignore the error correlation without effecting performance. Consequently, any *d*-dimensional problem can be treated as *d* independent scalar problems. Hence, for our numerical investigation, we will use data generated from the following scalar function that gives true states *Y* and noisy observations *P*:(11)$$\begin{array}{ccc} Y_{t} & = & 25+10\sin\frac{t}{15}+\sqrt{\alpha }\chi_{t},\\ P_{t} & = & Y_{t}+\sqrt{\beta }\epsilon_{t}, \end{array}$$
where 0≤t≤n, the observation’s noise $\epsilon_{t}∼N\left(0,1\right)$ with $\epsilon_{t}$ a white noise process, $\chi_{t}$ is an independent cumulative white noise process, $\chi_{t}=\sum_{i=1}^{t}{\hat{\xi }}_{i}\;\mathrm{where}\;{\hat{\xi }}_{i}∼N\left(0,1\right)$, α is the variance of the cumulative white noise and β is the variance of the observational noise.

We generated data with n initial length of n=500 using β=9 and α=1. From the dataset, we randomly deleted data points using a uniform distribution (cf. Figure 8a) so that the final dataset had $n^{\prime }\approx 125$ data points. Again, we used our usual measures <E> and <e> estimated from N=100 datasets for η=1.3 or 19 (depending on the tracking objective). To incorporate the non-constant sampling time, we included the sampling times into the matrix Equation (A12). Our numerical investigation shows that for increasing levels of deletion, the error increased relative to the uniform sampling. Specifically, we found that <E>≈3.3 (η=1.3) and <e>≈1.5 (η=19). Overall, our filter was still able to find a good enough approximation, as can be seen in Figure 8b.

### 5.2. Multiple Bearing Observations: Singularities

In this section, we will study another situation where non-Cartesian observations are used. Consider the case when the target is tracked in the xy-plane, by the position s=(x,y), using bearing observations $q=(\theta ,\theta^{\prime })$ from two distinct reference points (a,b) and $\left(a^{\prime },b^{\prime }\right)$. The transformation *f* between the different coordinates is defined by: (12)$$\begin{array}{ccc} \left(x,y\right) & = & (a,b)+m(\cos\theta ,\sin\theta ), \end{array}$$
(13)$$\begin{array}{cc} = & \left(a^{\prime },b^{\prime }\right)+m^{\prime }\left(\cos\theta^{\prime },\sin\theta^{\prime }\right), \end{array}$$
where under most circumstances, there exist unique $m,m^{\prime }\in R$. Hence, if we have bearing observations, then we need to find the raw position estimates, and that requires calculating *m* and $m^{\prime }$. These can be estimated by solving the linear equations:$$\begin{array}{ccc} a+m\cos\theta & = & a^{\prime }+m^{\prime }\cos\theta^{\prime },\\ b+m\sin\theta & = & b^{\prime }+m^{\prime }\sin\theta^{\prime }, \end{array}$$
which are equivalent to:$$\begin{array}{ccc} m\cos\theta -m^{\prime }\cos\theta^{\prime } & = & a^{\prime }-a,\\ m\sin\theta -m^{\prime }\sin\theta^{\prime } & = & b^{\prime }-b, \end{array}$$
and these can be expressed in matrix form as follows:(14)$$\left(\begin{array}{cc} \cos\theta & -\cos\theta^{\prime }\\ \sin\theta & -\sin\theta^{\prime } \end{array}\right)\left(\begin{array}{c} m\\ m^{\prime } \end{array}\right)=\left(\begin{array}{c} a^{\prime }-a\\ b^{\prime }-b \end{array}\right).$$

Solving these linear equations enables us to find the raw position estimates from bearing observations using the transformation *f* [27].

In this situation, it is difficult to compute the Jacobian matrix directly from the transformation *f*. However, note that the inverse of the transformation *f* is given by:(15)$$\begin{array}{ccc} \theta & = & \arctan\frac{y-b}{x-a}, \end{array}$$
(16)$$\begin{array}{ccc} \theta^{\prime } & = & \arctan\frac{y-b^{\prime }}{x-a^{\prime }}. \end{array}$$

Consequently, the matrix *K*, the Jacobian matrix of $f^{-1}$, is easily computed as follows:$$K=\left(\begin{array}{cc} \frac{-(y-b)}{r^{2}} & \frac{\left(x-a\right)}{r^{2}}\\ \frac{-(y-b^{\prime })}{r^{\prime 2}} & \frac{\left(x-a^{\prime }\right)}{r^{\prime 2}} \end{array}\right),$$
where $r^{2}={\left(x-a\right)}^{2}+{\left(y-b\right)}^{2}$ and $r^{\prime 2}={\left(x-a^{\prime }\right)}^{2}+{\left(y-b^{\prime }\right)}^{2}$, which requires simple computations once an estimate of (x,y) is obtained. Therefore, under the assumption that both *r* and $r^{\prime }$ are not close to zero, the information matrix can be approximated using the matrix *K* as follows:$$\begin{array}{ccc} I_{s} & = & K^{T}I_{q}K\\ & = & \left(\begin{array}{cc} \frac{-(y-b)}{r^{2}} & \frac{-(y-b^{\prime })}{r^{\prime 2}}\\ \frac{\left(x-a\right)}{r^{2}} & \frac{\left(x-a^{\prime }\right)}{r^{\prime 2}} \end{array}\right)\left(\begin{array}{cc} \sigma_{\theta }^{-2} & 0\\ 0 & \sigma_{\theta^{\prime }}^{-2} \end{array}\right)\left(\begin{array}{cc} \frac{-(y-b)}{r^{2}} & \frac{\left(x-a\right)}{r^{2}}\\ \frac{-(y-b^{\prime })}{r^{\prime 2}} & \frac{\left(x-a^{\prime }\right)}{r^{\prime 2}} \end{array}\right), \end{array}$$
where $\sigma_{\theta }^{-2}$ is the variance of the observational error in the first bearing component and $\sigma_{\theta^{\prime }}^{-2}$ is the variance in the second bearing component. It follows that the information matrix is given by:$$I_{s}=\left(\begin{array}{cc} \frac{\sigma_{\theta }^{-2}{\left(y-b\right)}^{2}}{r^{4}}+\frac{\sigma_{\theta^{\prime }}^{-2}{\left(y-b^{\prime }\right)}^{2}}{r^{\prime 4}} & \frac{-\sigma_{\theta }^{-2}\left(x-a\right)\left(y-b\right)}{r^{4}}+\frac{-\sigma_{\theta^{\prime }}^{-2}\left(x-a^{\prime }\right)\left(y-b^{\prime }\right)}{r^{\prime 4}}\\ \frac{-\sigma_{\theta }^{-2}\left(x-a\right)\left(y-b\right)}{r^{4}}+\frac{-\sigma_{\theta^{\prime }}^{-2}\left(x-a^{\prime }\right)\left(y-b^{\prime }\right)}{r^{\prime 4}} & \frac{\sigma_{\theta }^{-2}{\left(x-a\right)}^{2}}{r^{4}}+\frac{\sigma_{\theta^{\prime }}^{-2}{\left(x-a^{\prime }\right)}^{2}}{r^{\prime 4}} \end{array}\right).$$

Given this information matrix, we can apply the shadowing filter to find an approximated trajectory.

A problem arises in situations when the sensors and the target are collinear; i.e., $\theta \approx \pm \theta^{\prime }$. In such situations, the linear Equations (14) are singular or badly conditioned. This can lead to raw position estimates far from their true positions. To solve this problem, we have three options: we can drop the bad observation, replace it with a forecasted position or take into account that the information in this observation is less reliable than in other observations. Since the shadowing filter is estimating a trajectory from a sequence of observations, generally there is no harm in either of these solutions. Although, dropping bad observations leads to a non–uniform time-gap between observations where the shadowing filter still works successfully, as verified in the above investigations. We have chosen here to give our observations different weights, according to their reliability. To do this, we scaled the information matrix by using the rcond command in MATLAB. This command calculates the one-norm estimate of the reciprocal condition number as returned by LAPACK. rcond was used on the matrix giving information on the angles:$$\left(\begin{array}{cc} \cos\theta & -\cos\theta^{\prime }\\ \sin\theta & -\sin\theta^{\prime } \end{array}\right).$$

The output of rcond varies between zero and one. For angles that lead to an ill-conditioned Equation (14), we get small weights, but good observations will get maximum weighting.

In the example shown in Figure 9 and [27], we tracked an object using bearing observations provided by two moving sensors. The target moved on a circular path (sin(t/25),cos(t/25) for 0≤t≤100). It moved in a clockwise direction, starting from the 12 o’clock position. The first sensor moved between (−3,3) and (3,1), and the other between (−3,−2) and (3,1). Both sensors had a constant speed. To generate the bearing observations, we transformed the true states of the object to the bearing coordinates using Equations (15) and (16). We added white noise to both components with variances $\alpha_{\theta }=\alpha_{\theta^{\prime }}=0.05$ to obtain the desired multiple bearing noisy observations. After that, we transformed these noisy observations to the Cartesian coordinates using the transformation *f* after computing *m* and $m^{\prime }$ from Equation (14). We applied the shadowing filter at η=1 on these observations using the scaled information matrices to deal with the bad observations. In Figure 9, we can also see the effect of the different weighting. Note that when the target was between the four and five o’clock position, it was directly between the sensors $\theta =\pm \theta^{\prime }$. This led to a very bad observation, as seen by the large error. Similarly, at the end of the trajectory, the object was almost directly behind both sensors, which also led to a poorly-conditioned observations. However, using the one-norm estimates as described above enabled us to determine good approximations.

### 5.3. Chaotic Trajectory: Lorenz Model

In the previous sections and examples, we investigated the performance of the tracking filter for deterministic trajectories in one- and two-dimensional spaces. The natural step from here is to consider the realistic case of a tracking such as the chaotic motions seen in nature in insects, fish or birds.

We now apply the tracking technique to the Lorenz model [36]:
(17a)$$\begin{array}{ccc} \frac{dx}{dt} & = & \Sigma (y-x) \end{array}$$
(17b)$$\begin{array}{ccc} \frac{dy}{dt} & = & x(\rho -z)-y \end{array}$$
(17c)$$\begin{array}{ccc} \frac{dz}{dt} & = & xy-Bz \end{array}$$

Using the standard Lorenz model and typical parameters ρ=28, Σ=10 and B=8/3, we initialized the system with $\left(x_{0},y_{0},z_{0}\right)=\left(0,1,1.05\right)$ and generated a trajectory containing 2501 data points for t=[0,25]. The data evolved as per Equation (17) and were measured with some uncertainty following the incorporation of noise with variance of β=1 to each component as per Equation (11).

The dynamics were tracked using the shadowing filter with η=0.1, which was the optimal value for the chosen noise variance as found in Section 4, and a sampling rate of 0.1. We considered uncorrelated errors and treated the system as per the scalar case outlined in Section 3.

Figure 10a–c shows the successful tracking of a target in each of the variables, while Figure 11 shows these details in a projection of the three-dimensional state space. We note that as the shadowing filter did not contain any information about the vector field, the noisy observations were projected down on to the attracting manifold of the Lorenz model. Therefore, our filter acted similar to the other shadowing filters, which had information regarding the vector field encoded in them [17]. Although there were some estimation error, one can see at the outer loops in Figure 11, we note that our filter estimated the true time series most of the time.

Up until this point, we have been concerned with forecasting. However, this is closely related to noise reduction, and the results of the previous section can be viewed in such a context [37,38].

We assumed that the clean time series of each variable, $\tilde{x},\tilde{y}$ and $\tilde{z}$, is in the presence of measurement noise and aimed to separate these from the observed noisy data in order to restore the structure of the system and improve the quality of predictions [38], i.e., $x_{n}=\tilde{x_{t}}+\sqrt{\beta }\epsilon_{t}$, where $\epsilon_{t}$ is defined as in Equation (11).

For measurement noise applied to Equation (17), the approximated trajectory (green triangles) of Figure 10 and Figure 11 is the reduced noise trajectory. Figure 11 illustrates how the noise (red line) has smeared the chaotic attractor and obscured its structure and how the chaotic dynamics of the clean (blue line) trajectory were successfully captured.

### 5.4. Real-World Multiple Object Application

As a final application, we present how the shadowing filter can be used to track multiple objects. We used two-dimensional data produced from images of a real flock of ducks moving on a water surface. The dataset was provided by the authors of [39]. We focussed on one of the data files where a turning event occurred. We consider 25 individuals observed at 30 time points. We tracked the trajectories using the positional observations and extracted the corresponding acceleration for each bird. The difference between this application and the tracking of pigeons illustrated in Figure 1 was the size of the flock. Here, we have a larger number of individuals, but a relatively short trajectory. The results are shown in Figure 12, where the circles are the observations and the estimated paths are given as solid lines. The filter accurately tracked the position of each individual. The acceleration of each bird is shown in the lower panels of the figure. Note that this additional information about each individual’s state can be used to forecast its future dynamics and could be used to infer that the forces dominate such a collective behaviour [13].

## 6. Conclusions

In this paper and our previous ones [11,12,13], a new class of shadowing filters, applicable to observational data of objects moving in more than one dimension, was introduced and investigated. Our investigations demonstrate that this approach is highly versatile. First of all, it may be easily adapted to different settings. For instance, it can be implemented to track a single moving particle, a system of multiple objects [13] or be extended to consider rigid bodies instead [11]. Secondly, it is computationally inexpensive in comparison to alternative methods [11]. Finally, it outperformed the established and widely-used Kalman and particle filters in situations where nonlinearities were present [11]. Our tracking technique is applicable to real data and enables us to minimise measurement errors and overcome device failures, as well as allows for a reconstruction of the full dynamical state space from limited position-only data [12,13].

In this paper, specifically, optimization of the filter quality was achieved for two tracking objectives, namely “tracking the path that the object travelled” and “knowing the recent position of the object”, by tuning the smoothing parameter such that the (task-specific) error was minimized. Numerical investigations indicated that our tracking method was sufficiently robust that the benefit of taking error correlation into account was negligible, allowing for many systems to be treated as per the scalar case. Furthermore, we demonstrated that this tracking method can be applied to irregularly-sampled real-world data without additional effort, a situation often encountered when measurement devices fail to log some measurement points. We highlight the ability of the shadowing filter tracking method to avoid singularities in data, another challenge encountered frequently in practice. The shadowing filter was successfully applied to noisy data from the Lorenz model, and we briefly commented on the relation of this approach to noise reduction algorithms. Finally, a real-world application of multiple object tracking was provided, as we tracked a flock of 25 ducks moving on the water surface.

Lightweight and inexpensive tracking devices are now available and have opened up a new area of tracking possibilities [34,35,40,41,42,43,44], resulting in new observational data, which enable the study of movement, of animals for example, with so far unreached precision. Given these advantages and the resulting big data, it is clear that the new generation of filters needs to be fast and efficient. The outlined benefits of our filter together with its clear optimization possibility make it, in our opinion, a very strong contender for such applications, and we are looking forward to seeing the progress in this area based on our filter.

## Figures and Tables

**Figure 1 sensors-19-00931-f001:**
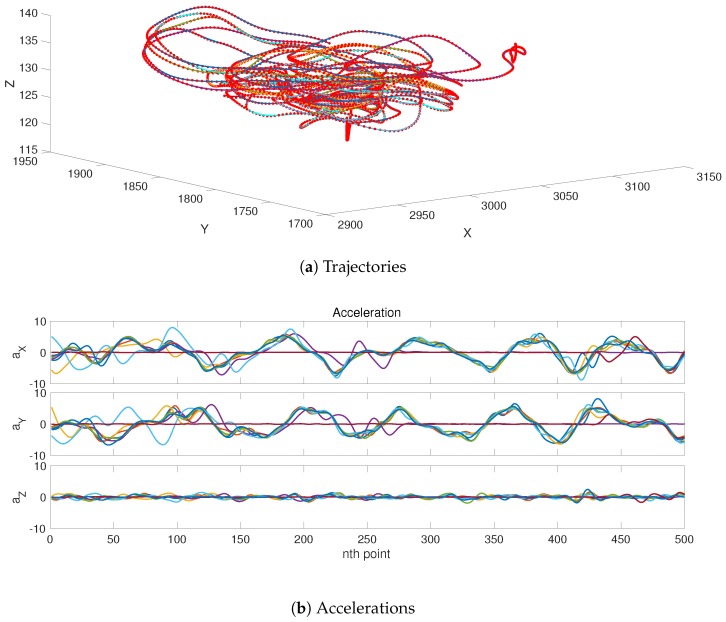
Shadowing filter applied to GPS data from a flock of pigeons. The shadowing filter successfully tracks the eight pigeons. In this demonstration, we refer to our previous works [12,13]. In (**a**), we track the trajectories of the eight pigeons (the coloured solid lines) from the GPS signals (red dots). In (**b**), we use the tracking algorithm to estimate the corresponding acceleration profiles for each pigeon (colours refer to different pigeons).

**Figure 2 sensors-19-00931-f002:**
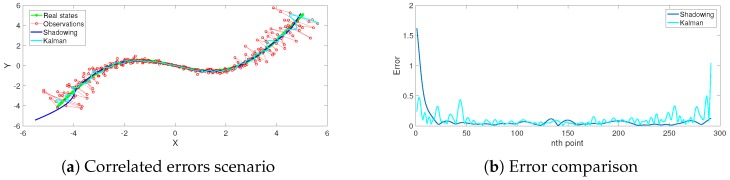
The superiority of the shadowing filter tracking methodology over a sequential filter methodology (Kalman filter). Error correlation has been taken into account in (**a**) and (**b**), while the singularity scenario is presented in (**c**) and (**d**). The shadowing filter shows obvious superiority, especially where error correlation increases the nonlinearity of the system far from the reference point at the origin in the first scenario and around the singular records in the second scenario. Note that the large error of the shadowing filter estimates at the beginning of the trajectory is expected, as explained in previous investigations [11,12].

**Figure 3 sensors-19-00931-f003:**
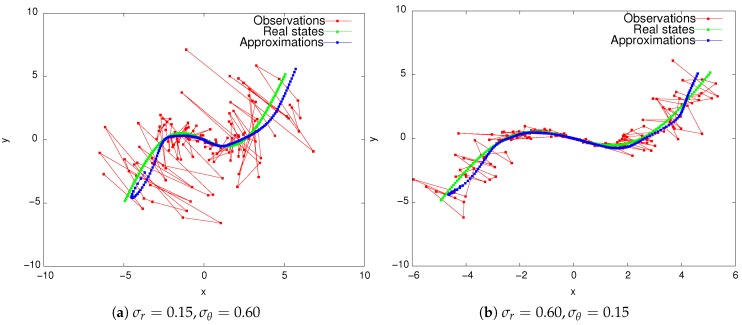
Taking error correlation into account when using range and bearing observations of different accuracies. The smoothing parameter is η=100 for both situations.

**Figure 4 sensors-19-00931-f004:**
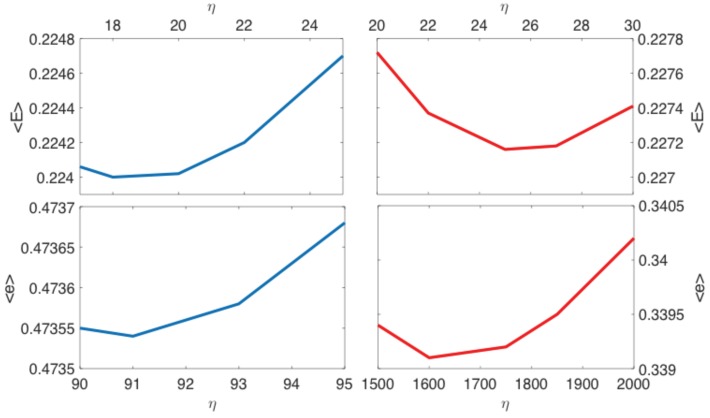
Optimisation of the smoothing parameter η: average root mean squared error 〈E〉 and average end-point error 〈e〉 as functions of the smoothing parameter η. We show both scenarios when the error’s correlation is taken into account (blue) or completely ignored (red). The minimum illustrates the existence of the optimal value of η that minimises the error and shows that incorporating the error’s correlation does not significantly improve the filter’s performance.

**Figure 5 sensors-19-00931-f005:**
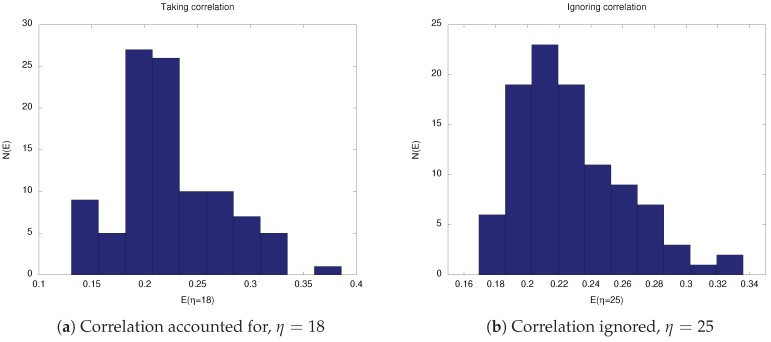
Histograms of the root mean squared error *E* from 100 observation sequences at the optimal value of η in both scenarios when the error’s correlation is taken into account or ignored.

**Figure 6 sensors-19-00931-f006:**
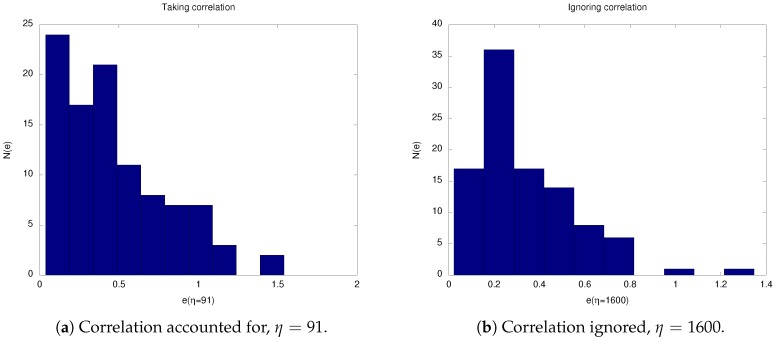
Histograms of the end-point error *e* from 100 observation sequences at the optimal value of η in both scenarios when error’s correlation is taken into account or ignored.

**Figure 7 sensors-19-00931-f007:**
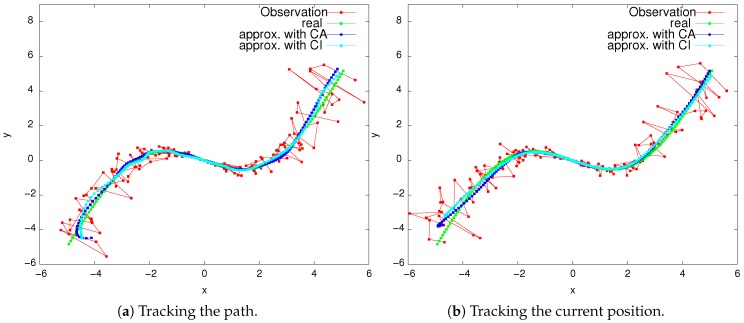
The filter performance in estimating (**a**) the path and (**b**) the current position, when error correlation is taken into account. Note that the true dynamics is given by Equation (5).

**Figure 8 sensors-19-00931-f008:**
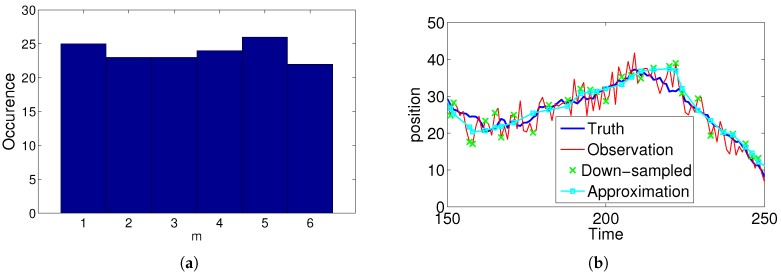
Performance of the filter when using non-uniform sampled data. (**a**) shows the histogram of sample times in the sequence, and (**b**) directly compares the observed and estimated trajectory, as well as the observed and non-uniformly down-sampled time series.

**Figure 9 sensors-19-00931-f009:**
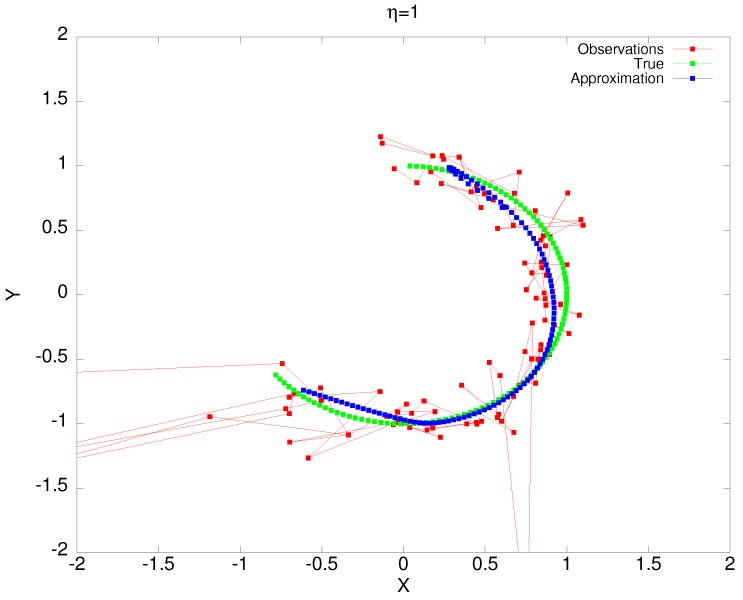
Performance of the shadowing filter in tracking an object using multiple bearing observations with η=1, $\alpha_{\theta }=\alpha_{\theta^{\prime }}=0.05$. Note how the filter is able to overcome the singularities in the observations and estimates the true dynamics quite well.

**Figure 10 sensors-19-00931-f010:**
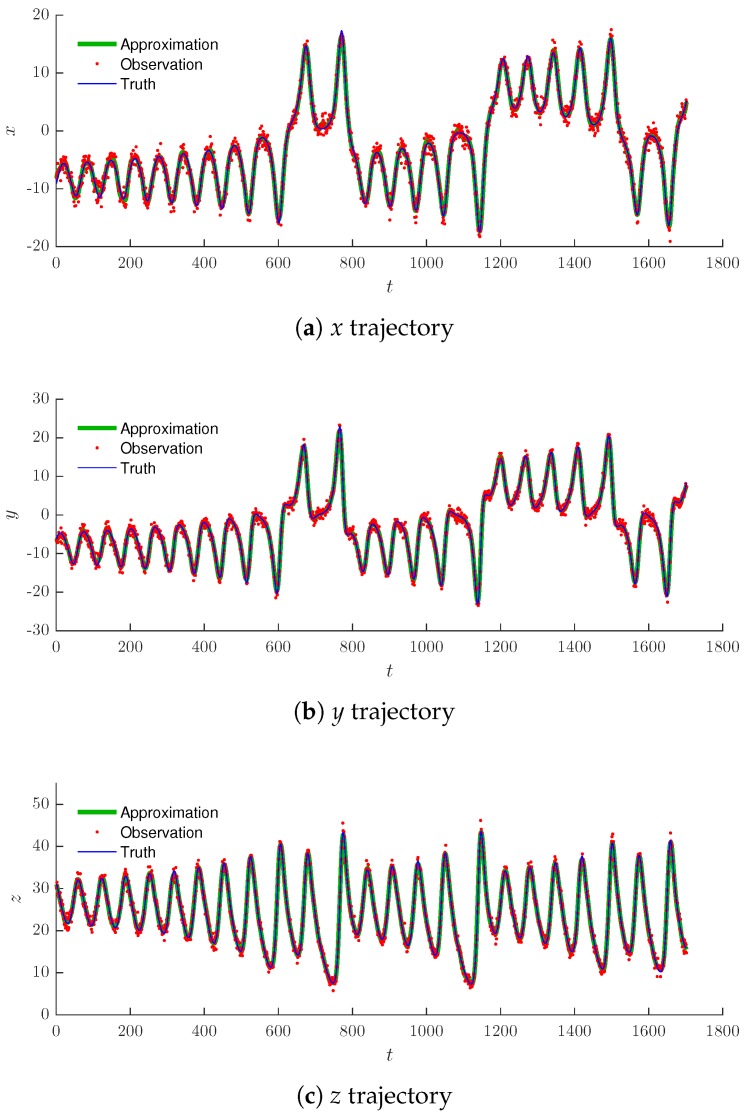
Projections onto each direction for the chaotic Lorenz model.

**Figure 11 sensors-19-00931-f011:**
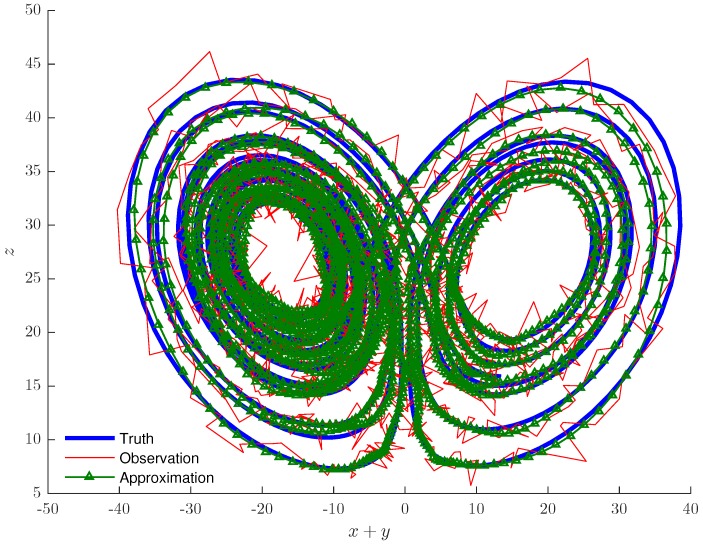
Chaotic Lorenz attractor (blue line), noisy observations (red line) and the tracking technique results (green triangles).

**Figure 12 sensors-19-00931-f012:**
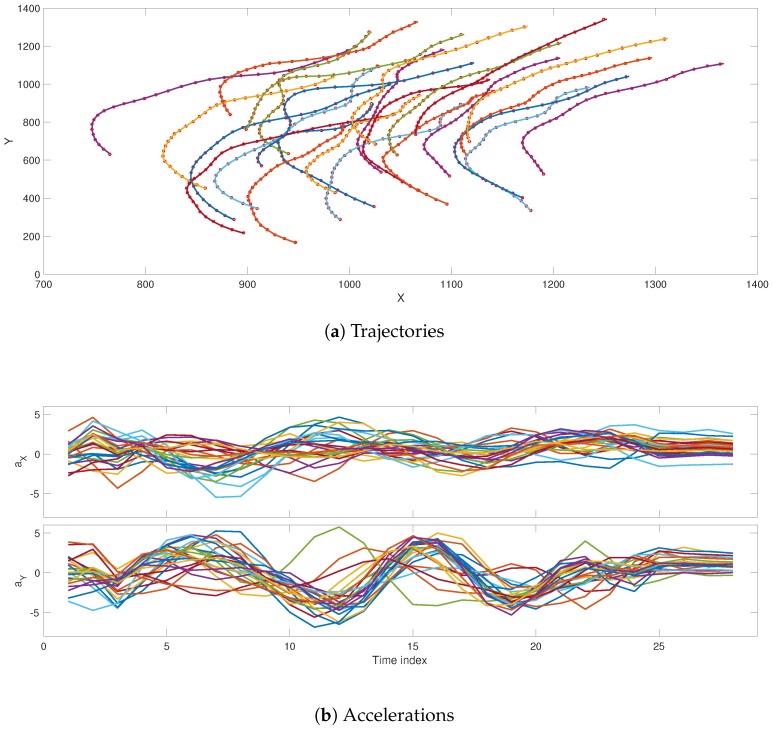
Shadowing filter applied to positional data from a group of ducks moving on the water surface. The shadowing filter successfully tracked the 25 individuals. In (**a**), we tracked the trajectories of the 25 ducks (the coloured solid lines) from the positional observations (red circles). In (**b**), we used the tracking algorithm to estimate the corresponding acceleration profiles for each duck (colours refer to different individuals).

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
