# Peer review of "Optimal Shadowing Filter for a Positioning and Tracking Methodology with Limited Information"

_sensors, 2019, doi:10.3390/s19040931_

Round 1

Reviewer 1 Report

The manuscript has investigated the performance of the so-called shadowing filter when dealing with correlated noise, uncorrelated noise, irregular sampling in time. Particular attention has been paid to the common bearing and range observation. The work is well motivated and well written. The shadowing filter is interesting and technically sound, although it is not so popular as compared with a (Bayesian) sequential filter such as the UKF and PF. However, there are some very relevant works that seemingly have been overlooked as addressed below, which if taken into consideration in the technical discussion and in the experimental/simulation comparison may lead to different conclusions. Overall, I recommend the acceptance of the manuscript after some revision.

First, as shown in (1a), the presented shadowing filter works with position-measurements (which may be directly observed or converted from nonlinear measurements such as bearing and range). Converted measurement has actually also been considered in the extensions of the classic Kalman filter, namely converted measurement filtering, as reviewed in Section 3.1 of the following paper [r1].

[r1] T.C. Li, et al, Approximate Gaussian conjugacy: parametric recursive filtering under nonlinearity, multimodality, uncertainty, and constraint, and Beyond, Front. Inform. Technol. Electron. Eng., 18(12):1913-1939, 2017.

It is actually known that, converted measurement will cause somehow loss of the information carried in the measurement, in addition to causing correlation or even bias (see the above reference), although it is obvious that linearizing the measurement makes it easy to solve the fitting problem as given in (1a). The more popular approach to using a filter or an estimator is working with the nonlinear measurements rather converting them. In view of this, the author may consider to compare their shadowing filter with a nonlinear filter such as the UKF/PF that works directly with the nonlinear measurements.  

Second, the formulation (1a) is actually a type of the classic weighted least square (WLS) estimator, which plus (1b-1d) can be categorised as constrained WLS. The solution to the linear WLS or constrained WLS is arguably well known; see, e.g., [r2, r3] both of which are influential.

[r2] L. El Ghaoui and H. Lebret, Robust Solutions to Least-Squares Problems with Uncertain Data, SIAM J. Matrix Anal. Appl., 18(4), 1035–1064.

[r3] S. Chandrasekaran, G. H. Golub, M. Gu, and A. H. Sayed, Parameter Estimation in the Presence of Bounded Data Uncertainties, SIAM J. Matrix Anal. Appl., 19(1), 235–252.

For nonlinear systems, there are also some popular WLS algorithms such as the two-stage weighted LS [r4] and linear correction least squares [r5].

[r4] Y. T. Chan and K. C. Ho, “A simple and efficient estimator for hyperbolic location,” IEEE Trans. Signal Processing, vol. 42, pp. 1905–1915, Aug. 1994.

[r5] Y. Huang, J. Benesty, G. W. Elko, and R. M. Mersereau, “Real-time passive source localization: A practical linear-correction least-squares approach,” IEEE Trans. Speech, Audio Processing, vol. 9, pp. 943–956, Nov. 2001.

In fact, with the development of nonlinear fitting and numerical approximation methods, the nonlinear WLS (and its constrained version) can be directly solved efficiently by using some of the off-the-shelf algorithms embedded in the popular software toolbox such as the MATLAB, see, e.g., [r6, r7]

[r6] T. Li, H. Chen, S. Sun, J.M. Corchado. Joint smoothing and tracking based on continuous-time target trajectory fitting, IEEE Trans. Auto. Sci. Eng. Online.

[r7] T. Li, "Single-Road-Constrained Positioning Based on Deterministic Trajectory Geometry," in IEEE Communications Letters, vol. 23, no. 1, pp. 80-83, Jan. 2019.

All of these WLS approaches and related seem more relevant to the shadowing filter than the Bayesian filters listed. In particular, the target trajectories studied look very smooth and suit the methods presented in [r6] or [r7] which also accommodate the correlated/uncorrelated noise and irregular sampling in time, although they do not take into account explicitly the velocity and acceleration. A comparison or at least a discussion is desired.

Finally, in lines 142-143 and somewhere else, the standard deviations of the range noise and the bearing noise have been directly compared, like sigma_r = 4 sigma_theta. This is incorrect/improper as they have different units in practice and cannot be directly compared. Please correct.  

Some minor comments

1. In equation (7), parentheses may be used to the content to be summarized over.

2. In figure 4, smaller intervals may be considered for testing for the best value of eta. Further, the size of the figure can be smaller to save space.

3. What is the meaning of the two sentences in lines 305 and 306?

4. Are tables 1-4 really necessary (as the results are already very clear in Figure 4)?

Author Response

All the comments and suggestions have been considered and appropriate modifications have been added to the manuscript and highlighted in red.

Reviewer 1:

The manuscript has investigated the performance of the so-called shadowing filter when dealing with correlated noise, uncorrelated noise, irregular sampling in time. Particular attention has been paid to the common bearing and range observation. The work is well motivated and well written. The shadowing filter is interesting and technically sound, although it is not so popular as compared with a (Bayesian) sequential filter such as the UKF and PF. However, there are some very relevant works that seemingly have been overlooked as addressed below, which if taken into consideration in the technical discussion and in the experimental/simulation comparison may lead to different conclusions. Overall, I recommend the acceptance of the manuscript after some revision.

We would like to thank Reviewer 1 for their effort in reviewing our study. In particular, we appreciate their drawing our attention to a number of additional references that have helped us have a more complete literature review.

First, as shown in (1a), the presented shadowing filter works with position-measurements (which may be directly observed or converted from nonlinear measurements such as bearing and range). Converted measurement has actually also been considered in the extensions of the classic Kalman filter, namely converted measurement filtering, as reviewed in Section 3.1 of the following paper [r1].

[r1] T.C. Li, et al, Approximate Gaussian conjugacy: parametric recursive filtering under nonlinearity, multimodality, uncertainty, and constraint, and Beyond, Front. Inform. Technol. Electron. Eng., 18(12):1913-1939, 2017.

It is actually known that, converted measurement will cause somehow loss of the information carried in the measurement, in addition to causing correlation or even bias (see the above reference), although it is obvious that linearizing the measurement makes it easy to solve the fitting problem as given in (1a). The more popular approach to using a filter or an estimator is working with the nonlinear measurements rather converting them. In view of this, the author may consider to compare their shadowing filter with a nonlinear filter such as the UKF/PF that works directly with the nonlinear measurements.

Thanks for your valuable comment. We had looked at the suggested reference and do see therelevanceto our work. We have added the reference to our introduction. Regarding the comparison with the other filters, we have done a detailed comparison with other methodsin:

Zaitouny A.A., Stemler T., Judd. K. [2017] “Tracking Rigid Bodies Using Only Position Data: A Shadowing Filter Approach Based on Newtonian Dynamics” Digital Signal Processing67, 81-90.

Judd, K. [2003] “Bayesian reconstruction of chaotic times series: Right results for the wrong reasons.” Physical Review E67, 026212. 

Judd K. and Stemler T [2009], “Failure of sequential bayesian filters and the success of shadowing filters in tracking of nonlinear deterministic and stochastic systems.” Phys. Rev. E 79, 066206.

Zaitouny A.A., Stemler T., Small M. [2017] “Tracking a Single Pigeon Using a Shadowing Filter Algorithm” Ecology and Evolution7(12), 4419-4431. 

Stemler, T. & Judd, K. [2009] “A guide to shadowing filters for forecasting and state estimation.” Physica D238, 1260-1273. 

Second, the formulation (1a) is actually a type of the classic weighted least square (WLS) estimator, which plus (1b-1d) can be categorised as constrained WLS. The solution to the linear WLS or constrained WLS is arguably well known; see, e.g., [r2, r3] both of which are influential.

[r2] L. El Ghaoui and H. Lebret, Robust Solutions to Least-Squares Problems with Uncertain Data, SIAM J. Matrix Anal. Appl., 18(4), 1035–1064.

[r3] S. Chandrasekaran, G. H. Golub, M. Gu, and A. H. Sayed, Parameter Estimation in the Presence of Bounded Data Uncertainties, SIAM J. Matrix Anal. Appl., 19(1), 235–252.

For nonlinear systems, there are also some popular WLS algorithms such as the two-stage weighted LS [r4] and linear correction least squares [r5].

[r4] Y. T. Chan and K. C. Ho, “A simple and efficient estimator for hyperbolic location,” IEEE Trans. Signal Processing, vol. 42, pp. 1905–1915, Aug. 1994.

[r5] Y. Huang, J. Benesty, G. W. Elko, and R. M. Mersereau, “Real-time passive source localization: A practical linear-correction least-squares approach,” IEEE Trans. Speech, Audio Processing, vol. 9, pp. 943–956, Nov. 2001.

In fact, with the development of nonlinear fitting and numerical approximation methods, the nonlinear WLS (and its constrained version) can be directly solved efficiently by using some of the off-the-shelf algorithms embedded in the popular software toolbox such as the MATLAB, see, e.g., [r6, r7]

[r6] T. Li, H. Chen, S. Sun, J.M. Corchado. Joint smoothing and tracking based on continuous-time target trajectory fitting, IEEE Trans. Auto. Sci. Eng. Online.

[r7] T. Li, "Single-Road-Constrained Positioning Based on Deterministic Trajectory Geometry," in IEEE Communications Letters, vol. 23, no. 1, pp. 80-83, Jan. 2019.

All of these WLS approaches and related seem more relevant to the shadowing filter than the Bayesian filters listed. In particular, the target trajectories studied look very smooth and suit the methods presented in [r6] or [r7] which also accommodate the correlated/uncorrelated noise and irregular sampling in time, although they do not take into account explicitly the velocity and acceleration. A comparison or at least a discussion is desired.

Thank you for bringing these to our attention, we have added a brief discussion in Section 2 related to these references.

Finally, in lines 142-143 and somewhere else, the standard deviations of the range noise and the bearing noise have been directly compared, like sigma_r = 4 sigma_theta. This is incorrect/improper as they have different units in practice and cannot be directly compared. Please correct.  

Corrected to avoid the direct comparison.

Some minor comments 

1. In equation (7), parentheses may be used to the content to be summarized over.

Modified as requested.

2. In figure 4, smaller intervals may be considered for testing for the best value of eta. Further, the size of the figure can be smaller to save space.

We have resized Figure 4. as requested. In these figures we demonstrate the existence of the optimal value (rather than exact determination) of the smoothing parameter that corresponds to minimal error. As demonstrated in the figure, the error’s convergence is very slow around the intervals, therefore we stopped our calculations at that level. 

3. What is the meaning of the two sentences in lines 305 and 306?

We agree that this was not clear and have modified the sentences to be more readable.

4. Are tables 1-4 really necessary (as the results are already very clear in Figure 4)?

In these tables, we demonstrate the process we have followed for investigation of the optimal value of the smoothing parameter. While, Figure 4 illustrates the final steps in the process, we feel it is helpful to add these calculations and tables into the appendix for interested readers.

Reviewer 2 Report

The paper is very interesting in my opinion. The filtering problem is very important in the field of signal processing. Generally, It is also well-written and well-structured. 

However, I have some suggestions to increase the quality of the paper and its impact. See below.

- It could be better to denote vectors with boldfaced letters.

- It is not clear to me, if the correlation matrix and/or the information matrix is estimated in advance (before the tracking algorithm is run). Please, clarify it.

- What is the observation model)considered in is Section 4? what is the perturbation noise?

- Please, improve the caption of Figures 4, 5 and 6 (giving more explanation). 

- It is important to improve the state-of-the-art to increase the impact and appealing of the paper.   

Section 2 about the Shadowing filters  is well-written.  However, I suggest to incorporate two more related references  of particle filtering:

 A. Doucet and A. M. Johansen. A tutorial on particle filtering and smoothing: fifteen years later. technical report, 2008. 

L. Martino, V. Elvira, G. Camps-Valls, "Group Importance Sampling for Particle Filtering and MCMC", Digital Signal Processing Volume 82, Pages 133-151, 2018.

In this two references above, the particle filtering approach is described in a more general way  (starting from “a sequential importance sampling with resampling” point of view) that allows to tackle more general models and applications. This way of introducing the particle filtering approach is more related to the  “shadowing filtering” that you are describing in Section 2. A proper literature can increase the number of interested readers and the possible impact of your work.

- Please, upload the final version of your manuscript in Arxiv and/or ResearchGate when/if published, to increase the diffusion and the possible citations of your works.

Author Response

All the comments and suggestions have been considered and appropriate modifications have been added to the manuscript and highlighted in red.

Reviewer 2:

The paper is very interesting in my opinion. The filtering problem is very important in the field of signal processing. Generally, It is also well-written and well-structured. 

However, I have some suggestions to increase the quality of the paper and its impact. See below.

We are grateful to reviewer 2 for their positive comments and thorough review and suggestions. The comments have greatly enhanced the presentation of our manuscript and helped to clarify a number of details and points made throughout.

- It could be better to denote vectors with boldfaced letters.

We agree and we have edited the manuscript by using boldfaced letters for vectors.

- It is not clear to me, if the correlation matrix and/or the information matrix is estimated in advance (before the tracking algorithm is run). Please, clarify it.

Yes, the information matrix has been estimated in advance using the observations and the corresponding variances by the equation of I_s in page 7.

We have included an additional sentence in page 7 when the matrices are first defined to clarify this point.

- What is the observation model considered in Section 4? what is the perturbation noise?

The observation model is introduced in equation (5) and we have now labelled it as such in the preceding sentence given by the simple, non-trivial observation model. The noise has been added with the equation inline145.

- Please, improve the caption of Figures 4, 5 and 6 (giving more explanation).

We have elaborated on and improved the captions. We have also expanded on the captions of Fig. 7, 8 and 9

 -  It is important to improve the state-of-the-art to increase the impact and appealing of the paper.

Section 2 about the Shadowing filters  is well-written.  However, I suggest to incorporate two more related references  of particle filtering:

A. Doucet and A. M. Johansen. A tutorial on particle filtering and smoothing: fifteen years later. technical report, 2008. 

L. Martino, V. Elvira, G. Camps-Valls, "Group Importance Sampling for Particle Filtering andMCMC", Digital Signal Processing Volume 82, Pages 133-151, 2018.

In this two references above, the particle filtering approach is described in a more general way (starting from “a sequential importance sampling with resampling” point of view) that allows to tackle more general models and applications. This way of introducing the particle filtering approach is more related to the “shadowing filtering” that you are describing in Section 2. A proper literature can increase the number of interested readers and the possible impact of your work.

The suggested papers are very relevant and have been incorporated as references into our paper in the introduction.

Regarding the comparison with the other filters and state-of-art, we have done this comparison in detail in:

Zaitouny A.A., Stemler T., Judd. K. [2017] “Tracking Rigid Bodies Using Only Position Data: A Shadowing Filter Approach Based on Newtonian Dynamics” Digital Signal Processing,67, 81-90.

Judd, K. [2003] “Bayesian reconstruction of chaotic times series: Right results for the wrong reasons.” Physical Review E67, 026212. 

Judd K. and Stemler T [2009], “Failure of sequential bayesian filters and the success of shadowing filters in tracking of nonlinear deterministic and stochastic systems.” Phys. Rev. E 79, 066206.

Zaitouny A.A., Stemler T., Small M. [2017] “Tracking a Single Pigeon Using a Shadowing Filter Algorithm” Ecology and Evolution,7(12), 4419-4431. 

Stemler, T. & Judd, K. [2009] “A guide to shadowing filters for forecasting and state estimation.” Physica D238, 1260-1273.

- Please, upload the final version of your manuscript in Arxiv and/or ResearchGate when/if published, to increase the diffusion and the possible citations of your works.

We will upload the final version in ResearchGate and Arxiv.

Reviewer 3 Report

The manuscript presents an experimental evaluation of shadowing filter, which was designed to track moving objects. The results are promising and several interesting observations are reported regarding the tracking accuracy and robustness of the filter. The paper is well written and easy to follow. However, the manuscript should be revised according to the following comments before being published.

1. The shadowing filter has been already presented in three previous papers published by the Authors. The novelty of the results presented in current manuscript should be better explained in the context of the previous works.

 2. The main experiments were performed for synthetic datasets (trajectories) only. Just one real-world example related to pigeon's flight is presented. However, it comes from previously published paper. The accuracy and robustness of the filter could be proven using additional real-world datasets.

3. The comparison with the alternative tracking methods (Kalman, particle filters) should be extended. It would be useful to show how the alternative methods perform in the tracking examples that are presented in Figs. 3, 7, 8, 9, 10, and 11.  Appropriate discussion should be provided regarding results that can be obtained by using the alternative methods. It should be explained if the alternative methods are not applicable in particular tracking examples.

Author Response

All the comments and suggestions have been considered and appropriate modifications have been added to the manuscript and highlighted in red.

Reviewer 3:

The manuscript presents an experimental evaluation of shadowing filter, which was designed to track moving objects. The results are promising and several interesting observations are reported regarding the tracking accuracy and robustness of the filter. The paper is well written and easy to follow. However, the manuscript should be revised according to the following comments before being published.

We would like to thank Reviewer 3 for their review of our study and for the thoughtful suggestions, which have helped to expand our analysis and discussions.

1. The shadowing filter has been already presented in three previous papers published by the Authors. The novelty of the results presented in current manuscript should be better explained in the context of the previous works.

We have edited the introduction to incorporate your comment and better explain what is novel in this paper. This was also stated in the conclusion.

 2. The main experiments were performed for synthetic datasets (trajectories) only. Just one real-world example related to pigeon's flight is presented. However, it comes from previously published paper. The accuracy and robustness of the filter could be proven using additional real-world datasets.

We added to the manuscript a new real-world problem in a new section 5.4.

3. The comparison with the alternative tracking methods (Kalman, particle filters) should be extended. It would be useful to show how the alternative methods perform in the tracking examples that are presented in Figs. 3, 7, 8, 9, 10, and 11.  Appropriate discussion should be provided regarding results that can be obtained by using the alternative methods. It should be explained if the alternative methods are not applicable in particular tracking examples.

While we understand the need for comparisons, the focus of this paper is more on the filter’s merits, namely, its performance when dealing with correlated/uncorrelated noise, irregular sampling in time, singularities and multiple object tracking. In addition, we provide some discussion on sequential filters in section 2 where we reference our previous investigations focussing on the comparison. Moreover Fig. 2 provides some comparison that supports our main conclusion on the shortcomings of such sequential filters.

Regarding the comparison with the other filters, we have done this comparison in details in:

Zaitouny A.A., Stemler T., Judd. K. [2017] “Tracking Rigid Bodies Using Only Position Data: A Shadowing Filter Approach Based on Newtonian Dynamics” Digital Signal Processing67, 81-90.

Judd, K. [2003] “Bayesian reconstruction of chaotic times series: Right results for the wrong reasons.” Physical Review E67, 026212. 

Judd K. and Stemler T [2009], “Failure of sequential bayesian filters and the success of shadowing filters in tracking of nonlinear deterministic and stochastic systems.” Phys. Rev. E 79, 066206.

Zaitouny A.A., Stemler T., Small M. [2017] “Tracking a Single Pigeon Using a Shadowing Filter Algorithm” Ecology and Evolution7(12), 4419-4431. 

Stemler, T. & Judd, K. [2009] “A guide to shadowing filters for forecasting and state estimation.” Physica D238, 1260-1273.

Round 2

Reviewer 3 Report

The manuscript has been improved and now is suitable for publication in Sensors.